# Acute Eosinophilic Pneumonia Complicated with Venous Thromboembolic Disease—Diagnostic and Therapeutic Considerations

**DOI:** 10.3390/diagnostics12061425

**Published:** 2022-06-09

**Authors:** Ewa Jankowska, Iwona Bartoszuk, Katarzyna Lewandowska, Małgorzata Dybowska, Lucyna Opoka, Witold Tomkowski, Monika Szturmowicz

**Affiliations:** 11st Department of Lung Diseases, National Research Institute of Tuberculosis and Lung Diseases, 01-138 Warsaw, Poland; esrubka@gmail.com (E.J.); i.bartoszuk@igichp.edu.pl (I.B.); dybowska@mp.pl (M.D.); w.tomkowski@igichp.edu.pl (W.T.); monika.szturmowicz@gmail.com (M.S.); 2Department of Radiology, National Research Institute of Tuberculosis and Lung Diseases, 01-138 Warsaw, Poland; lucyna.opoka@gmail.com

**Keywords:** acute eosinophilic pneumonia, pulmonary embolism, venous thrombosis, thrombophilia, antithrombin deficiency

## Abstract

Acute Eosinophilic Pneumonia (AEP) is a rare idiopathic disease caused by an accumulation of eosinophils in the pulmonary alveoli and interstitial tissue of the lungs. The onset of symptoms is acute; some patients develop respiratory failure. The diagnosis is based on clinical symptoms, diffuse interstitial infiltrates in the lungs on imaging studies, and eosinophilia exceeding 25% on bronchoalveolar lavage or pleural fluid smear. Smokers are primarily at increased risk for the disease. We present a case of venous thromboembolic disease (VTE) that developed in the course of AEP in a previously healthy male smoker. Complete remission of the disease was achieved with anticoagulation therapy combined with a low dose of steroids. Surprisingly, further diagnostics revealed the presence of thrombophilia: antithrombin (AT) deficiency and increased homocysteine level. According to our knowledge, this is the first case of VTE diagnosed in the course of AEP combined with thrombophilia.

## 1. Introduction

Acute eosinophilic pneumonia (AEP) is a rare interstitial lung disease, with an incidence rate of 9/100,000 cases [1]. It affects mostly young males, 20–40 years of age [1]. Smokers are at increased risk of the disease. However, AEP may also develop in the persons exposed to other inhalants, including psychostimulants [2]. AEP has also been described in the course of COVID-19 disease [3,4]. Most AEP patients present with profound hypoxemia, often requiring oxygen therapy and, occasionally, mechanical ventilation [1]. Despite the severe disease course, the treatment results with corticosteroids are excellent [1]. Clinical relapses are seldom observed after corticosteroids withdrawal [1]. Blood and pulmonary eosinophilia are considered risk factors for VTE [5,6]. Nevertheless, the clinical data concerning VTE in the course of AEP are missing. We present AEP complicated with acute VTE, diagnosed in previously healthy, young male smoker. 

## 2. Case Report

A 53-year-old male smoker (10 pack years) was admitted to the Infectious Diseases University Hospital in September 2020 due to acute respiratory disease presenting with fever, dry cough, exertional dyspnea, and pleuritic chest pain. Chest X-ray revealed bilateral infiltrates localized in lower lung fields and a small amount of fluid in both pleural cavities. A PCR test for SARS-CoV-2 was negative. The patient was diagnosed with bilateral pleuropneumonia. Amoxicillin with clavulanic acid was implemented as empirical treatment, and the patient was transferred to the National Tuberculosis and Lung Diseases Research Institute for further therapy. On admission, he remained in poor performance status, with a fever of 38–39 °C, dyspnea, and chest pain. Physical examination revealed blood pressure 140/60 mmHg, heart rate 105/min, SaO_2_—93%, and bilateral lower limb edema, more pronounced on the left side. On auscultation, diminished respiratory sounds were noted in the base of both lungs with few crepitations in this region. Laboratory analysis showed elevated C-reactive protein (CRP) 81 mg/L (N < 5 mg/L) and markedly increased D-dimers concentration (25,000 ng/mL; N < 500 ng/mL). Leucocyte count was 13.8 × 10^9^/L, with eosinophilia 3.07 × 10^9^/L (22%), procalcitonin concentration was within a normal range—0,12 ng/mL (N < 0.5 ng/mL). Capillary blood gas analysis showed hypoxemia (PaO_2_—62 mmHg) and hypocapnia (PaCO_2_—32 mmHg). Chest X-ray revealed the progression of interstitial infiltrates and an increase in left-sided pleural fluid volume (Figure 1). CT pulmonary angiography (CTPA) was performed due to high suspicion of pulmonary embolism. Multiple emboli localized in the distal part of the main right pulmonary artery (PA), superior lobe PA, intermediate lobe PA, and inferior lobe PA have been found (Figure 2a). On the left side, the emboli were localized in the lingula artery. In addition, parenchymal infiltrates were described in the right upper lobe, lower lobes, and pleural effusion bilaterally (Figure 2b). Doppler ultrasonography revealed massive deep vein thrombosis of the left lower limb. In the common femoral vein, thrombi comprised up to 90% of the lumen, thrombotic lesions propagated to the iliac vein, no flow was observed in the femoral and popliteal veins, and thrombotic lesions propagated distally to the veins of the soleus muscle and the tibial veins. Echocardiography did not reveal signs of right ventricular dysfunction, tricuspid regurgitation gradient (TRG) was 34 mmHg, and right ventricular outflow Doppler acceleration time (AcT)—90 ms. There was no significant left ventricular disorder (left ventricular ejection fraction—56%), and no valve pathology was noted. NT-proBNP concentration was 200 pg/mL (N < 125 pg/mL). Low-risk PE (simplified version Pulmonary Embolism Severity Index—0 points) [7] in the course of left-sided proximal deep veins’ thrombosis was diagnosed, and anticoagulation with low molecular weight heparin (LMWH) was started (enoxaparin 1 mg/kg, twice daily). 

Blood cultures were negative. Thus, empiric treatment with ceftriaxone and levofloxacin was started. Clinical improvement was not achieved. Persistent febrile episodes, hypoxemia, increased CRP, and eosinophilia persisted. Control CTPA revealed the regression of PA emboli but the progression of left-sided pleural effusion. Thoracentesis performed under ultrasound guidance yielded 350 mL of bloody exudate with 66% of eosinophils and 24% of lymphocytes. Cultures and cytology of pleural fluid were negative. Immunoglobulin E (IgE) concentration was 52 IU/mL (N: 0–100 IU/mL), anti-neutrophil cytoplasmic antibodies (ANCA), and antinuclear antibodies (ANA) were negative.

Based on clinical and radiological presentation combined with systemic and pleural fluid eosinophilia exceeding 25%, acute eosinophilic pneumonia (AEP) was diagnosed. Steroid therapy was started with prednisone at an initial dose of 20 mg per day. In the next few days, the patient’s performance status improved significantly. Laboratory tests showed a decrease in CRP to 28 mg/L and eosinophilia to 0.57 × 10^9^/L (8.7%)—Table 1. Respiratory failure resolved.

Chest X-ray revealed regression of pleural fluid and parenchymal consolidations. Antibiotics and oxygen therapy were discontinued. The patient was discharged from the hospital with a recommendation to continue steroid therapy and anticoagulant therapy at home. Four months later, the patient did not report any respiratory complaints, CRP, blood cell count, and D-dimers were within normal limits (Table 1). Chest X-ray showed complete remission of lung and pleural disease (Figure 3). CTPA revealed complete resolution of PA emboli, pleural fluid, and lung consolidations (Figure 4a,b). Doppler ultrasonography showed complete recanalization of the common femoral vein, organizing thrombi comprising up to 40% of the femoral vein and 40–75% of the popliteal vein. 

Prednisone dose was gradually decreased and stopped. Anticoagulation was continued, and LMWH was substituted with rivaroxaban 20 mg/day. Further diagnostics for congenital thrombophilia was planned in 3–6 months. It revealed AT deficiency and increased homocysteine level (>50.0 µmol/L). Reduced antithrombin activity (anti-IIa 76.85%, anti-Xa 80.99%) suggested a type 2 antithrombin deficiency. Therefore, lifelong therapy with rivaroxaban was proposed.

## 3. Discussion

The presented case report illustrates difficulties concerning the complex pathogenesis of VTE in real-life patients. The initial diagnosis of pulmonary embolism and deep vein thrombosis of the left lower limb was made, and subsequent antithrombotic treatment resulted in the resolution of the PA emboli. Still, it did not affect pleural fluid and lung consolidations. It was hypothesized that there was another cause for the patient’s clinical deterioration. Because of the increasing amount of fluid in the left pleural cavity, thoracentesis was performed. It documented the presence of exudate with 66% of eosinophils. The results led to the recognition of pulmonary eosinophilia. After excluding collagen tissue disease and vasculitis, the final diagnosis was AEP.

Recently published modified diagnostic criteria of AEP include: 1. Acute febrile illness of less than one month; 2. Hypoxemia; 3. Bilateral diffuse pulmonary infiltrates on chest radiography, bilateral ground-glass attenuations, and consolidations, as well as an interlobular septal thickening in chest computed tomography; 4. Pulmonary eosinophilia: eosinophils >25% in BAL or pleural fluid, infiltration with eosinophils diagnosed on lung biopsy; 5. Absence of all known causes of eosinophilic lung disease [1]. In this case, AEP was diagnosed based on clinical symptoms (exertional dyspnea, dry cough, fever), interstitial infiltrates in chest X-ray and chest CT scan, and a high percentage of eosinophils in the pleural fluid. Blood eosinophilia, documented in the presented case, is not the leading clinical sign of AEP. It is diagnosed in 30–40% of patients only and may be delayed [1,8]. In eosinophil-associated diseases, the C-reactive protein level usually is low [9]. In our case, it was significantly elevated due to VTE.

According to the literature, tobacco smokers constitute the vast majority of patients with acute eosinophilic pneumonia [1,8,10]. The presented patient had a history of long-term smoking. According to available sources, acute eosinophilic pneumonia often occurs either a short time after starting to smoke [1,11] or when the number of cigarettes increases [1]. An acute hypersensitivity reaction to inhaled antigens present in tobacco smoke is suspected of playing a role in the pathogenesis of the disease [1]. Damage to the airway epithelium leads to the secretion of cytokines and chemokines, which then stimulate the activation and recruitment of eosinophils in lung tissue [1,12]. There are also reports of AEP in patients who give up traditional cigarettes but use e-cigarettes, tobacco products based on heating tobacco, water pipe, and marijuana [2]. The presented patient denied using non-cigarette smoking products. Before his illness, he did not take any medication regularly.

The most interesting problem concerning the presented patient was the possible impact of AEP on the coagulation system, resulting in acute VTE development. The data included in the literature confirm that both systemic and pulmonary eosinophilia results in enhanced coagulation mechanisms [6,13,14]. Activated eosinophils eject mitochondrial DNA and release granules’ contents, among others—eosinophil peroxidase (EP) and eosinophil cationic protein (ECP), with subsequent forming the eosinophil extracellular traps (EETs) [13]. EETs formation was confirmed in BALF of patients with chronic eosinophilic pneumonia [15] and Hodgkin lymphoma [16]. Hypercoagulability may result from the direct pro-coagulant properties of EP, as well as from the platelet-activating actions of proteins present in eosinophilic granules [6,13]. ECP enhances factor XII-dependent reactions, shortening the coagulation time [5]. It is also postulated that major basic protein (MBP) inhibits the ability of thrombomodulin to bind to thrombin and activate protein C, an anticoagulant [14]. Thus, the balance between pro-coagulant and anticoagulant factors in eosinophilic diseases is disturbed, leading to hypercoagulability. Venous thromboembolism associated with tissue and/or systemic eosinophilia was previously reported in eosinophilic esophagitis [17], eosinophilic granulomatosis with polyangiitis (EGPA) [18], food protein-induced allergic proctocolitis [19], and idiopathic hyper-eosinophilic syndrome [20,21,22], nevertheless it was not reported in AEP. 

The treatment of AEP is based on corticosteroids. In severe respiratory failure, therapy is initiated with methylprednisolone 60–125 mg administered intravenously every 6 hours [8], and once the clinical condition is stabilized, treatment is switched to the oral form. Most patients need supplemental oxygen, sometimes even mechanical ventilation [8]. In milder cases, oral corticosteroids can be given from the beginning at a dose of 0.5–1 mg/kg body weight, progressively tapered. The presented patient received a lower dose of prednisone than recommended due to a mild AEP course and to minimize the risk of corticosteroid side effects in the context of coexisting acute VTE. The optimal treatment duration of AEP has not been established yet, but published data indicate that even a 2-week treatment may be sufficient [1,23]. Treatment with low-dose prednisone was prolonged to 6 months in the presented patient due to the possible relationship between pulmonary eosinophilia and massive anatomic VTE. Complete resolution of PA emboli, as well as lung consolidations and pleural effusion, were observed in the course of combination therapy with LMWH and steroids. Signs of deep-vein thrombosis diminished as well, but organized thrombi still comprised 40–75% of femoral and popliteal veins. Prednisone was stopped after 6 months, and enoxaparin was substituted with rivaroxaban. Anticoagulation treatment was planned until the moment thrombophilia would be excluded. Surprisingly, the patient was diagnosed with AT deficiency and increased homocysteine level.

AT deficiency is a rare coagulation disorder, found in 0.5% of the population and 4-7% of unprovoked VTE. Still, it is one of the strongest risk factors for VTE, increasing its’ probability by 50 times compared to the average population [24,25]. The literature data indicate that AT deficiency may result in VTE, especially in additional risk factor occurrences, such as surgery, trauma, or immobilization [24]. In the presented patient, medical history was negative concerning VTE episodes until 53 years of age. Therefore, it may be speculated that the development of VTE episode was provoked by the combination of risk factors, such as tissue and blood eosinophilia in the course of AEP and hereditary thrombophilia. Based on the actual guidelines, testing for thrombophilia is indicated in patients with VTE episode below 50 years of age, in case of unprovoked VTE episode, recurrent VTE episodes, or in those with atypical thrombus localization [7]. These indications were not fulfilled in the presented patient. Still, he was qualified for thrombophilia testing due to the massive anatomical character of VTE and uncertain AEP’s role in its pathogenesis. Lifelong anticoagulation with direct oral anticoagulant (DOAC) was proposed based on the obtained results.

## 4. Conclusions

Acute eosinophilic pneumonia is a rare condition with a good prognosis related to steroid treatment. In some patients, thromboembolic complications may occur in the course of the disease. However, additional risk factors of thromboembolic disease, i.e., inherited thrombophilia, should always be considered. 

## Figures and Tables

**Figure 1 diagnostics-12-01425-f001:**
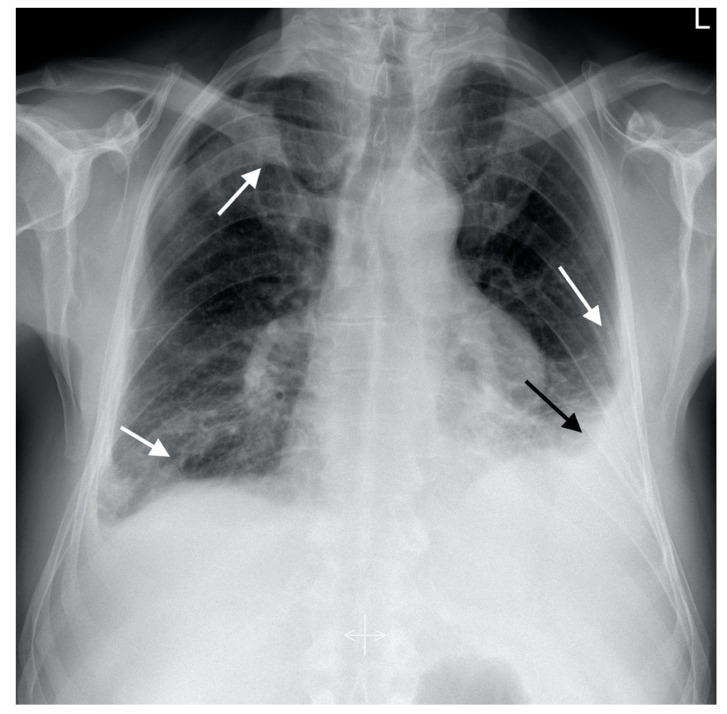
Postero-anterior chest X-ray showing bilateral interstitial infiltrates (white arrows) and left-sided pleural effusion (black arrow).

**Figure 2 diagnostics-12-01425-f002:**
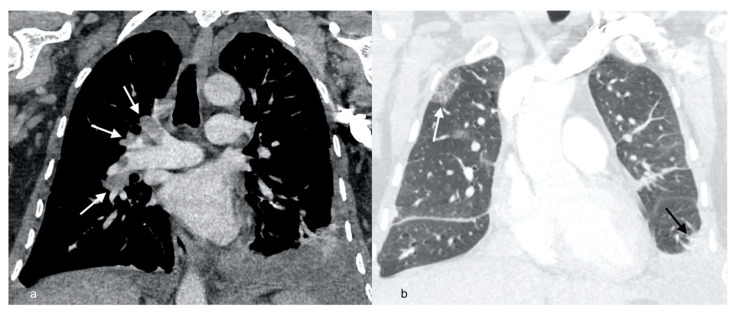
Computed tomography pulmonary angiography (CTPA) scan showing massive right pulmonary artery embolism (white arrows) (**a**), high resolution computed tomography (HRCT) of the chest showing ground glass opacities in the right upper lobe (white arrow), and left-sided pleural effusion (black arrow) (**b**).

**Figure 3 diagnostics-12-01425-f003:**
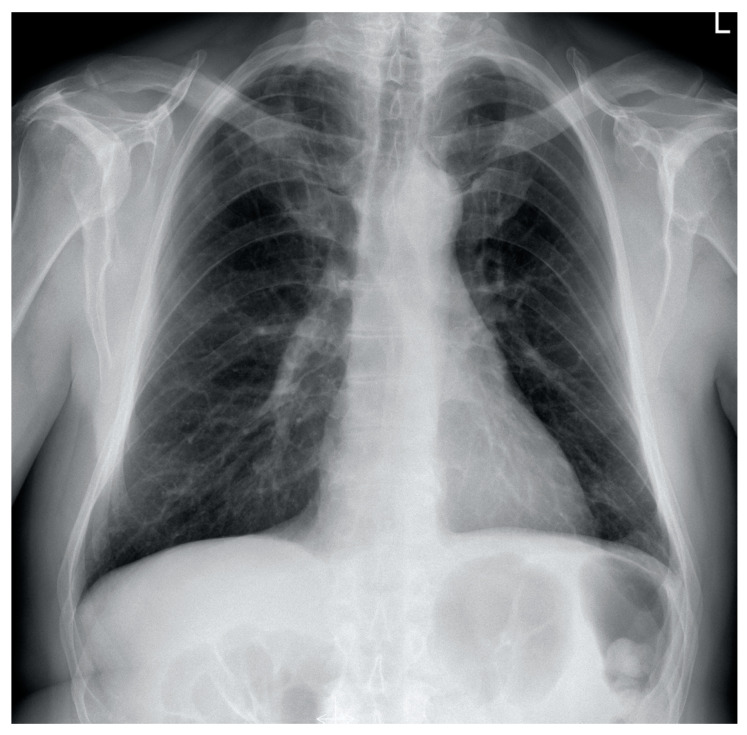
Postero-anterior chest X-ray showing complete regression of interstitial infiltrates and pleural effusion.

**Figure 4 diagnostics-12-01425-f004:**
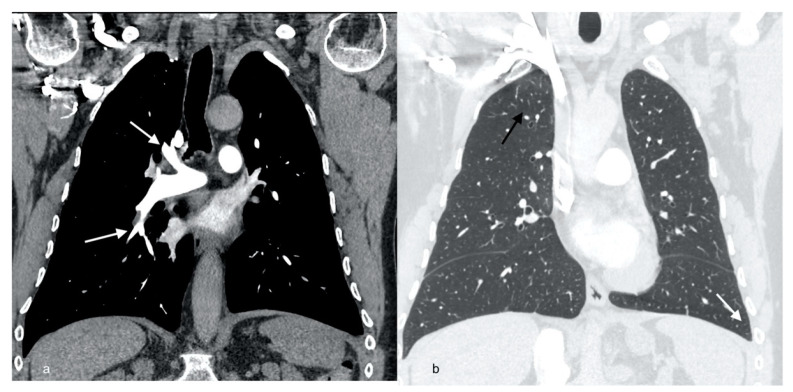
Computed tomography pulmonary angiography (CTPA) scan showing complete resolution of right pulmonary artery embolism (arrows) (**a**), high resolution computed tomography (HRCT) of the chest showing resolution of ground glass opacities in the right upper lobe) black arrow) and left-sided pleural effusion (white arrow) (**b**).

**Table 1 diagnostics-12-01425-t001:** Patient’s blood and imaging tests on admission, on discharge, and after four months follow-up.

	On Admission	On Discharge	4 Months Follow-Up	Reference Range
**Body temperature (°C)**	38–39	Normal	Normal	Normal
**White blood cell count (WBC) (×10^9^/L)**	13.8	6.53	6.65	3.98–10.04
**Eosinophil count (×10^9^/L)**	3.07	0.57	0.14	0.04–0.36
**Eosinophils (%)**	22	8.7	2.1	0.7–5.8
**C-reactive protein** **(CRP) (mg/L)**	81	28	<5	<5
**D-dimer (ng/mL)**	25,491	7567	176	<500
**PaO_2_ (mmHg)**	62	71.5	102.2	65–90
**PaCO_2_ (mmHg)**	32	37.1	41.6	35–45
**Radiological picture**	Pulmonary embolism, parenchymal infiltrates, bilateral pleural fluid	Regression of PA thrombi, pleural fluid and parenchymal consolidations	Complete resolution of PA thrombi, pleural fluid and lung consolidations	
**Doppler ultrasonography**	Massive venous thrombosis of left lower limb		Complete recanalization of femoral vein, organizing thrombi comprising up to 40% of femoral vein and 40–75% of popliteal vein	
**Echocardiography**	Mild pulmonary hypertension (TRG 34 mmHg, AcT 90 ms)		No evidence of pulmonary hypertension (TRG 25 mmHg)	

## Data Availability

Not applicable.

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
