# Peer review of "Acute Eosinophilic Pneumonia Complicated with Venous Thromboembolic Disease—Diagnostic and Therapeutic Considerations"

_diagnostics, 2022, doi:10.3390/diagnostics12061425_

Round 1

Reviewer 1 Report

Jankowska and colleagues report on a case of acute eosinophilic pneumonia complicated with venous thromboembolic disease and subsequent diagnosis of AT deficiency.

The manuscript is well written and the topic of eosinophil-related VTE deserves to be brought to the forefront.

Nevertheless, the manuscript could be improved :

- the discussion could be better structured. As it stands, it aggregates many small paragraphs whose connections is sometimes not so obvious.

- it should be mentionned that blood eosinophilia can be absent or delayed in acute eosinophilic pneumonia

- The reference PMID: 22599359 which implemented short-treatment durations (ie 2 weeks) should be cited.

- The authors could discuss the fact that C-reactive protein levels tend to be low in eosinophil-associated diseases, unless confounding factor such as eosinophil-related myocarditis, pleural infusion or thrombosis (PMID: 30317003): contrasting with mild pulmonary damage, the high CRP level measured in the patient warranted seeking for VTE.

- More data are needed on AT deficiency, homocystein level and MTHR mutation status which are only briefly listed.

Author Response

Dear Reviewer,

Thank you for a thorough revision of our manuscript entitled “Acute Eosinophilic Pneumonia Complicated with Venous Thromboembolic Disease – Diagnostic and Therapeutic Considerations”.

We revised our manuscript and implemented most of the suggestions.

  1. the discussion could be better structured. As it stands, it aggregates many small paragraphs whose connections is sometimes not so obvious

The paragraphs in the discussion were compressed and linked closely.

  1. It should be mentionned that blood eosinophilia can be absent or delayed in acute eosinophilic pneumonia

The information that only 30-40% of patients develop eosinophilia in peripheral blood is included in the discussion (Line 142). We added information that blood eosinophilia can be delayed in AEP (Line 142).

  1. The reference PMID: 22599359 which implemented short-treatment durations (ie 2 weeks) should be cited

This citation has been included (Line 184).

  1. The authors could discuss the fact that C-reactive protein levels tend to be low in eosinophil-associated diseases, unless confounding factor such as eosinophil-related myocarditis, pleural infusion or thrombosis (PMID: 30317003): contrasting with mild pulmonary damage, the high CRP level measured in the patient warranted seeking for VTE.

This information has been added (Line 143-144).

  1. More data are needed on AT deficiency, homocystein level and MTHR mutation status which are only briefly listed.

Unfortunately, we do not have complete patient records. Nevertheless, some information about test results has been added (Line 114-116).

Reviewer 2 Report

This is a very interesting and rare case of Acute Eosinophilic Pneumonia complicated with Thromboembolic disease, which has been treated successfully with prednisolone and LMWH. However, patient required to use DOAC for life-long due to AT deficiency was also found in this episode. I think it will provide more useful information and knowledge to readers after the group revised the manuscript.

# Introduction:

  1. There are 4 paragraphs in the introduction. Suggest try to condense it into one paragraph.
  2. I think the diagnostic criteria of AEP could be placed into the case report part or discussion part.
  3. What is the most unique part in this case? AEP comorbid with VTE and AT deficiency was diagnosed after that. If this part is your unique, try to link it into the end of introduction.

#. Case reports:

  1. This part also has the same problems as the introduction part. That is: too many paragraphs. Please try to condense it.
  2. The risk factors of AEP should mention the personal history not only smoking, but also vaping, E- cigarette, illicit drug use, drug exposure (e.g, antimicrobial, antidepressant…). Suggest adding more information about past and personal history.

#. Discussion:

  1. This part also has the same problems as the introduction part. That is: too many paragraphs. Please try to condense it.
  2. From Line159-178: this part should be merged to explain the possible mechanism of AEP induced VTE.
  3. I think AT deficiency accidentally found in this patient is an important point in this case report, as your title “Diagnostic and Therapeutic Consideration”. So, merge this information into one paragraph (from Line 198 to 213).
  4. Suggest to draw a picture regarding “how to evaluate AEP with VTE step-by-step” to fit the scope and name of the submitted journal “Diagnostics”.

Author Response

Dear Reviewer,

Thank you for a comprehensive revision of our manuscript entitled “Acute Eosinophilic Pneumonia Complicated with Venous Thromboembolic Disease – Diagnostic and Therapeutic
Considerations”.

We revised our manuscript according to your suggestions.

  1. Introduction:

There are 4 paragraphs in the introduction. Suggest try to condense it into one paragraph. I think the diagnostic criteria of AEP could be placed into the case report part or discussion part. What is the most unique part in this case? AEP comorbid with VTE and AT deficiency was diagnosed after that. If this part is your unique, try to link it into the end of introduction.

The manuscript has been corrected. The introduction now has one paragraph (Line 28-38).

  1. Case reports:

2.1. This part also has the same problems as the introduction part. That is: too many paragraphs. Please try to condense it.

It has been corrected. We tried to condense paragraphs.

2.2. The risk factors of AEP should mention the personal history not only smoking, but also vaping, E- cigarette, illicit drug use, drug exposure (e.g, antimicrobial, antidepressant…). Suggest adding more information about past and personal history.

Some information about the patient’s history is added to the manuscript (Line 154-156).

  1. Discussion:

3.1 This part also has the same problems as the introduction part. That is: too many paragraphs. Please try to condense it.

It has been corrected. Now there are fewer paragraphs.

3.2. From Line159-178: this part should be merged to explain the possible mechanism of AEP induced VTE.

This part is merged now to explain the possible mechanism of AEP-induced VTE (Line 157-174).

3.3. I think AT deficiency accidentally found in this patient is an important point in this case report, as your title “Diagnostic and Therapeutic Consideration”. So, merge this information into one paragraph (from Line 198 to 213).

It has been corrected. This information is now in one paragraph (Line 193-207)

3.4. Suggest to draw a picture regarding “how to evaluate AEP with VTE step-by-step” to fit the scope and name of the submitted journal “Diagnostics”.

In our opinion, the relationship between AEP and VTE has not been thoroughly examined yet. Such cases are extremely rare.  We concluded that preparing the graph on how to diagnose these two entities together could be misleading, and we decided not to do this.

Round 2

Reviewer 1 Report

The manuscript is significantly improved. All queries have been adressed. I have no further comments.

Reviewer 2 Report

The author has revised the paper appropriately.